# Exogenous Hormone Factors in Relation to the Risk of Malignant Melanoma in Women: A Systematic Review and Meta-Analysis

**DOI:** 10.3390/cancers14133192

**Published:** 2022-06-29

**Authors:** Manuela Chiavarini, Giulia Naldini, Irene Giacchetta, Roberto Fabiani

**Affiliations:** 1Department of Experimental Medicine, Section of Public Heath, University of Perugia, 06129 Perugia, Italy; manuela.chiavarini@unipg.it; 2School of Specialization in Hygiene and Preventive Medicine, University of Perugia, 06129 Perugia, Italy; naldini.giulia@gmail.com; 3Department of Chemistry, Biology and Biotechnology, University of Perugia, 06129 Perugia, Italy; roberto.fabiani@unipg.it

**Keywords:** oral contraceptive, hormone replace therapy, melanoma, meta-analysis

## Abstract

**Simple Summary:**

Many epidemiological studies have examined the relationship between cutaneous malignant melanoma (CMM) and both endogenous oestrogen exposure (e.g., age at menarche and parity) and exogenous hormone use (e.g., oral contraceptives (OCs) and menopausal hormone therapy (MHT)). Though a previous meta-analysis investigating the relationship between characteristics of female endocrine status and CMM risk found no significant association, the potential role of THERAPY AS oral contraceptive (OC) and hormonal replacement therapy (MHT) use still remains controversial. Since then, several studies have been published about the therapy with contrasting results, while CMM incidence continues to increase with a significant gender divergence. The therapy of OC and MHT may play a role in CMM and the removal of this could be useful as emerging therapeutics in melanoma. Therefore, we conducted this systematic review and meta-analysis to summarize the evidence and derive a more accurate estimation of exogenous hormone factors in women and CMM.

**Abstract:**

The influence of exogenous female hormones on the risk of developing malignant melanoma in women remains controversial. The aim of our review and meta-analysis is to summarize the evidence and derive a more accurate estimation of the association between oral contraceptives (OCs) or menopausal hormone therapy (MHT) and the risk of developing malignant melanoma in women. PubMed, Web of Science, and Scopus database were searched for studies published up until October 2021. The PRISMA statement and MOOSE guidelines were followed. Studies were pooled using a random effects model. Heterogeneity was explored with the chi-square-based Cochran’s Q statistic and the I^2^ statistic. Publication bias was assessed with Begg’s test and Egger’s test. Forty-six studies met the eligibility criteria. The pooled analysis (26 studies) on OC use and the risk of developing cutaneous malignant melanoma (CMM) showed no significant association, but demonstrated significant association for cohort studies (OR 1.08, 95% CI 1.01–1.16; I^2^ = 0.00%, *p* = 0.544). The pooled analysis (16 studies) showed a significantly increased risk of CMM in association with MHT (OR 1.15, 95% CI 1.08–1.23; I^2^ = 25.32%, *p* = 0.169). Stratifying the results by study design showed that a significant increased risk of CMM was associated with MHT in the cohort studies (OR 1.12; 95% CI 1.04–1.19; I^2^ = 0%, *p* = 0.467). No significant publication bias could be detected. Further studies are needed to investigate the potential association with formulation, duration of use, and dosage of use, and to better understand the role of possible confounders.

## 1. Introduction

Cutaneous malignant melanoma (CMM) is the sixth most common cancer in women worldwide [1], with an estimated 137,000 (129,800–144,600) new cases in 2018 [2] and represents the 16th cause of cancer death [1].

A considerable decline in mortality rates was observed in the period 2013–2017 (6.3% per year), while incidence rates increased by 1.9% (1.5–2.2) per year [1].

Ultraviolet radiation exposure [3,4]; naevi (common and atypical) count [5]; freckle density; phenotypic characteristics (skin type I, skin color, eye color, and hair color) [6]; a family history of melanoma [6,7]; and familiar susceptibility due to low-, medium-, or high-penetrance genes [8] are well-established risk factors for CMM. Intriguingly, several pieces of epidemiological data have noted a significant gender divergence in CMM incidence [1,2,9]. Particularly, the incidence of CMM is higher in adolescent and young adult females [10]. Compared to males, the probability of developing CMM increases in women under the age of 50, but is lower at an older age [9]. Moreover, a gender difference in CMM survival has been noted for the early stage of the disease, though results are controversial in more advanced stages [11,12].

Following the observation of these sex differences, many epidemiological studies have examined the relationship between CMM and both endogenous estrogen exposure (e.g., age at menarche and parity) and exogenous hormone use (e.g., oral contraceptives [OCs] and menopausal hormone therapy [MHT]) [13,14,15,16,17,18,19,20,21,22,23,24,25,26,27,28,29,30,31,32,33,34,35,36,37,38,39,40,41,42,43,44,45,46,47,48,49,50,51,52,53,54,55,56]. Though a previous meta-analysis [57] investigating the relationship between characteristics of female endocrine status and CMM risk found no significant association, the potential role of oral contraceptive (OC) and hormonal replacement therapy (MHT) use remains controversial.

Since then, several studies have been published with contrasting results. Therefore, we conducted this systematic review and meta-analysis to summarize the evidence and derive a more accurate estimation of malignant melanoma risk and exogenous hormone factors in women.

## 2. Materials and Methods

This systematic review and meta-analysis were conducted and reported according to the meta-analysis of observational studies in epidemiology (MOOSE) guidelines [58] and the preferred reporting items for systematic reviews and meta-analyses (PRISMA) statement [59].

### 2.1. Search Strategy and Data Source

We carried out a comprehensive literature search, without restrictions, up until 1 October 2021 through PubMed (http://www.ncbi.nlm.nih.gov/pubmed/, accessed on 28 April 2022), Web of Science (http://apps.webofknowledge.com, accessed on 28 April 2022), and Scopus (https://www.scopus.com/, accessed on 28 April 2022), databases to identify all the original articles investigating the association between exogenous hormone use and malignant melanoma risk in women. The following search medical subject headings (MeSH) and key words were used: (“oral contraceptive” OR “exogenous hormones” OR “hormonal therapy” OR “hormone therapy”) AND (melanoma OR “skin cancer”). In addition, the reference lists of included articles and recent relevant reviews were manually examined to identify additional relevant publications.

### 2.2. Eligibility Criteria

Publications were eligible if they: (i) evaluated the relationship between exogenous hormone use and malignant melanoma in women; (ii) used a case–control, prospective, or cross-sectional study design; (iii) presented risk estimates (odds ratio, OR; relative risk, RR; or hazard ratio, HR) with 95% confidence intervals (CIs). In the presence of several publications from the same study, the publication with the biggest sample was selected. For each potentially included study, two investigators independently conducted the selection, data abstraction, and quality assessment. Disagreements were resolved by discussion or in consultation with a third author. Although it is useful to have background information, reviews and meta-analyses were excluded. No studies were excluded based on weakness of design or data quality.

### 2.3. Data Extraction and Quality Assessment

From the included studies, we extracted the following information: the first author’s last name, the year of publication, country, the study design, the sample size (when possible, the number of cases and controls and incident cases, as well cohort size), population characteristics (age, ethnicity), the duration of follow-up for cohort studies, tumor characteristics (CMM; superficial spreading melanoma, SSM; nodular melanoma, NM; and uveal/intraocular melanoma), the identification of cases, exposure assessment, OCs exposure (the duration of use, the time since the most recent OC use, the time since the first OC use, the status of OCs, and the age at first use), MHT exposure (the duration of use, the status of MHT use, regimen, the type of MHT, and the route of administration), risk estimates with 95% CIs for the different categories of exogenous hormone use, a *p*-value for trend, and adjustment of confounding factors. When multiple estimates were reported in the article, those adjusted for the most confounding factors were pulled out. The Newcastle–Ottawa Scale (NOS) [60] was used for the quality evaluation of the enrolled studies. NOS adopted a star system, with a total score ranging from 0 to 9. A total score of ≥7 indicated a high-quality study. Two investigators individually performed the quality evaluation of each selected study and disagreements were settled by a joint reevaluation of the original article with a third author.

### 2.4. Statistical Analysis

We evaluated the association between exogenous hormone use (OCs and MHT) and malignant melanoma’s risk in women using the statistical program ProMeta version 3.0 (IDo Statistics-Internovi, Cesena, Italy). For the overall estimation, the relative risk and hazard ratio were taken as an approximation to the OR, and the meta-analysis was performed as if all types of ratio were ORs. The combined risk estimate was calculated using a random effect model.

The chi-square-based Cochran’s Q statistic and the I2 statistic were used to evaluate heterogeneity in results across studies [61]. The I2 statistic yields results ranged from 0% to 100% (I2 = 0–25%, no heterogeneity; I2 = 25–50%, moderate heterogeneity; I2 = 50–75%, large heterogeneity; and I2 = 75–100%, extreme heterogeneity) [62]. Results of the meta-analysis may be biased if the probability of publication is dependent on the study results. We used the method by Begg and Mazumdar [63] and the method by Egger et al. [64] to detect publication bias. Both methods were tested for funnel plot asymmetry—the former was based on the rank correlation between the effect estimates and their sampling variances, and the latter was based on a linear regression of a standard normal deviate on its precision. If a potential bias was detected, we further conducted a sensitivity analysis to assess the robustness of combined effect estimates, and the possible influence of the bias, and to have the bias corrected. We also conducted a sensitivity analysis to investigate the influence of a single study on the overall risk estimate, by omitting one study in each turn. We considered the funnel plot to be asymmetrical, if the intercept of Egger’s regression line deviated from zero, with a *p*-value < 0.05.

## 3. Results

### 3.1. Study Selection

The study selection process is shown in Figure 1. The primary literature research through PubMed (*n* = 179), Web of Science (*n* = 364), and Scopus (*n* = 915) databases returned a total of 1458 records. Duplicates (*n* = 767) were removed. Based on the title and abstract revision, we identified 54 eligible records on exogenous hormone use and malignant melanoma in women. Hand searching of reference lists of both already selected articles and recent relevant reviews led to the identification of no additional item. Of the 54 records subjected to full-text revision, 8 were further excluded because they failed to meet the inclusion criteria (1 did not report malignant melanoma as an outcome, 2 studies did not report exposure for OCs or MHT, 1 used men as a reference group, and 5 reported no risk estimates).

Therefore, at the end of the selection process, 46 studies were eligible for final inclusion in the systematic review and meta-analysis. Of these, 15 studies reported risk estimation of both OCs and MHT for malignant melanoma in women. Thirty-nine records investigated the relationship between OCs and malignant melanoma in women and twenty-two records investigated the relationship between MHT and malignant melanoma in women.

### 3.2. Meta-Analysis on the Risk of Developing Malignant Melanoma and OC Use

#### 3.2.1. Study Characteristics and Quality Assessment

The detailed characteristics of the studies on the association between OCs and malignant melanoma are shown in Table A1. Among the 39 selected studies [13,15,16,17,20,22,23,24,25,26,27,28,29,30,31,32,33,35,36,37,38,39,40,41,42,43,44,45,46,47,49,50,51,52,53,54,55,56,65], 25 are case–control studies [13,15,16,17,27,28,29,32,33,35,36,37,38,39,41,42,43,44,45,46,49,53,54,55,56] and 14 are cohort studies [20,22,23,24,25,26,30,31,40,47,50,51,52,65].

The evaluated outcomes in this meta-analysis were CMM, superficial spreading melanoma (SSM), nodular melanoma (NM), and uveal/intraocular melanoma. Thirty-five studies [13,15,17,22,23,24,25,26,27,28,29,30,31,33,35,36,38,39,40,41,42,43,44,45,46,47,49,50,51,52,53,54,55,56,65] investigated the risk of CMM associated with OCs, 9 studies [22,27,28,35,36,38,39,46,55] investigated the risk of SSM associated with OCs, and 7 studies [22,27,28,35,38,39,46] investigated the risk of NM associated with OCs. Three studies [16,32,37] analyzed the risk of uveal/intraocular melanoma associated with OCs and one study (20) analyzed the risk of melanoma associated with OCs. Regarding the association between CMM and OCs, 26 studies [13,15,17,22,23,27,28,29,30,31,33,35,36,38,39,40,42,44,45,47,50,51,52,54,55,56] included cases of in situ and invasive melanomas, whereas 7 studies [24,25,26,41,43,49] selected invasive melanomas only. The study by Palmer et al. [46] referred to severe invasive cutaneous melanoma. Thirty-four studies [13,15,17,22,23,24,25,26,27,28,29,30,31,33,35,36,38,39,40,41,42,43,44,45,46,47,49,50,51,52,53,54,55,56] reported risk estimates for SSM, three studies [16,32,37] for uveal/intraocular melanoma, and one study [20] for melanoma. Sixteen studies [13,20,22,23,24,31,35,36,38,45,47,53,54,55,56,65] assessed the outcome through record linkage to cancer registries, thirteen studies [17,26,27,28,29,32,33,39,41,42,43,44,49] assessed the outcome through histology and/or pathology confirmation, two studies [16,25] collected outcome information from general practitioners (GP) records, five studies [37,46,50,51,52] collected outcome information from the hospital records, one study [15] collected outcome information from either a pathology report or hospital discharge notes, and one study [40] did not specify the source of information. The study by Hannaford et al. [30] collected outcome information from GP records in one cohort and from hospital discharge record in the other one. Twenty-three studies [17,27,28,29,32,33,35,36,37,38,39,42,43,44,45,46,49,50,51,52,55,65] assessed OCs exposure through an interview, eight studies [15,16,22,23,24,25,26,53] through the administration of a questionnaire, and two studies [13,30] through either a questionnaire or an interview, while two studies [13,31] collected information from GP or medical records and one study [47] from pharmacy records, and the study by Koomen et al. [41] extracted data from the national registry. No information on exposure assessment was available in the study by Kay et al. [40]. Nine studies [22,25,29,31,38,42,46,51,52] reported risk estimates related to the time since the most recent OC use, seven studies [22,23,25,26,42,46,49] to the age at the first use, five studies [22,25,26,46,50] to the status of OC use, and four studies [25,38,46,65] to the time since the first use. Twenty studies [13,15,17,25,26,27,28,29,30,31,32,33,36,37,40,45,46,50,51,52] reported risk estimates as RR, thirteen studies [16,35,38,39,41,42,43,44,49,53,54,55,56] as OR, and five studies [20,22,23,24,65] as HR, whereas one study [47] reported SIR. One study [24] referred to never-users for all cases that have never used OCs or that have used OCs for less than a year.

The study-specific quality scores of selected studies are shown in the last column on the right of Table A1. The quality scores ranged from 0 to 8 (median: 6; mean: 6.1). The median values of cohort studies and case–control studies were seven and six, respectively. Among cohort studies, ten records [20,22,23,24,30,31,47,51,52,65] had a high score, three [25,26,30] had a medium score, and one study [40] had a low score. Eight case–control studies [35,37,41,43,44,45,49,54] had a high score, sixteen case–control studies [13,15,16,27,28,29,32,33,36,38,39,42,46,53,55,56] had a medium score, and one [17] had a low score.

#### 3.2.2. Meta-Analysis

Twenty-six studies [13,15,20,22,23,24,25,26,29,31,33,37,39,40,41,42,43,44,45,49,52,53,54,55,56,65] included in the systematic review were used for the overall risk estimation of CMM (Table 1, Figure 2a). One study [47] was excluded as reporting SIR and no risk estimates. In the overall analysis, OC use did not significantly affect the risk of developing CMM. Stratifying the results by study design, the time since the most recent OC use, and status of use showed no significant association between the risk of developing CMM and OC use. The stratification by study design showed a significant association for cohort studies; the stratification by age at the first use showed a significant association for an age greater than 20 years old. Stratifying the analysis by melanoma morphology showed that OC use did not significantly affect the risk of developing SSM or NM. In the overall analysis, the risk of developing uveal/intraocular melanoma showed no significant association with OC use.

#### 3.2.3. Sensitivity Analyses

Sensitivity analyses suggested that the estimates were slightly modified by any single study. In particular, a small change was found in the risk estimates after removing the study by Koomen et al. [41] (OR: 1.05; 95% CI: 0.99, 1.13; *p* = 0.239). However, removing the study by Østerlind et al. [45] resulted in a small increment of melanoma risk, which became statistically significant (OR: 1.09; 95% CI: 1.02, 1.16; *p* = 0.008).

#### 3.2.4. Publication Bias

No significant publication bias was detected with Egger’s or Begg’s tests (Table 1, Figure A1).

### 3.3. Meta-Analysis on the Risk of Malignant Melanoma and MHT Use

#### 3.3.1. Study Characteristics and Quality Assessment

The detailed characteristics of the studies on the association between MHT and malignant melanoma are shown in Table A2.

Among the 22 selected studies [14,16,17,18,19,20,21,23,24,26,32,34,35,36,37,41,43,44,45,48,65,66], 11 are case–control studies [16,17,32,34,35,36,37,41,43,44,45] and 11 are cohort studies [14,18,19,20,21,23,24,26,48,65,66].

The evaluated outcomes in this meta-analysis were CMM and uveal/intraocular melanoma. Eighteen studies [14,17,18,19,21,23,24,26,34,35,36,41,43,44,45,48,65,66] investigated the risk of CMM associated with MHT, three studies [16,32,37] investigated the risk of uveal/intraocular melanoma associated with MHT, and one study [20] investigated the risk of melanoma with MHT. Regarding the association between the risk of CMM and MHT, 10 studies [14,17,18,21,23,35,36,44,45,48] included cases of in situ and invasive melanomas, whereas 6 studies [19,24,26,34,41,43] selected invasive melanomas only. Three studies reported risk estimates for SSM [21,35,36], two studies [21,35] reported risk estimates for NM, and one study [21] reported risk estimates for LMM and ALM. Fourteen studies [14,18,19,20,21,23,24,34,35,36,45,48,65,66] assessed the outcome with record linkage to cancer registries, six studies [17,26,37,41,43,44] assessed the outcome with histology and/or pathology confirmation, one study [16] collected outcome information from general practitioners’ records, and one study [32] collected outcome information from the ocular oncology unit. Ten studies [16,17,32,35,36,37,43,45,65,66] assessed MHT exposure with an interview, six studies [14,21,23,24,26,44] with the administration of a questionnaire, and one study [20] with a questionnaire and medical records, whereas five studies [18,34,41,48] used a national registry or a database of drug prescriptions, and one study [19] used the medical reimbursement register of the national social insurance. Eleven studies [19,21,23,24,32,35,36,37,41,45,65] reported risk estimates for the duration of MHT use, four studies [18,21,23,66] for the status of MHT use, two studies [18,23] for regimen therapy, seven studies [14,18,21,23,45,48,66] for the type of MHT, and three studies [21,23,34] for the route of administration. Nine studies [14,17,18,26,32,36,37,45] reported risk estimates as RR, six studies [16,34,35,41,43,44] as OR, six studies [20,21,23,24,65,66] as HR, and two studies [19,48] reported SIR. One study [34] used as reference a category named non-users, which included patients who did not use MHT (excluding intravaginal estrogens) in the five years prior to diagnosis and one year after diagnosis.

Table A2 shows study-specific quality scores of the selected studies. The quality scores ranged from 3 to 9 (median: 7; mean: 6.3). The median value for both cohort studies and case–control studies was seven. Among cohort studies, eight records [18,19,20,21,23,24,65,66] had a high score and three [14,26,48] had a medium score. Six case–control studies [34,35,41,43,44,45] had a high score, four case–control studies [16,32,36,37] had a medium score, and one [17] had a low score.

#### 3.3.2. Meta-Analysis

Sixteen studies [14,17,18,20,21,23,24,26,34,35,41,43,44,45,65,66] included in the systematic review were used for the overall risk estimation of CMM (Table 2, Figure 2b).

Two studies [19,48] were excluded as reporting SIR and no risk estimates. We found that the risk of developing CMM was significantly higher in ever-users of MHT (OR 1.15, 95% CI 1.08–1.23). Stratifying the results by study design showed a significantly increased risk of CMM in cohort studies only (OR 1.12, 95% CI 1.04–1.19). Current MHT users had a significant higher risk (+19%) of CMM. Stratifying the analysis for the route of administration showed a significantly increased risk of CMM for both oral administration (OR 1.19, 95% CI 1.11–1.27) and, more noticeably, transdermal–cutaneous administration (OR 1.36, 95% CI 1.19–1.54). Stratifying the results by the type of MHT showed a significant positive association with the risk of developing CMM of ET only (OR 1.34, 95% CI 1.18–1.52). No significant association with the duration of MHT use was found.

Three studies [18,21,23] included in the systematic review were selected for the overall risk estimation of uveal/intraocular malignant melanoma. MHT use showed no significant association with the risk of developing uveal/intraocular malignant melanoma.

#### 3.3.3. Sensitivity Analyses

Sensitivity analyses investigating the influence of a single study on the CMM risk estimates suggested that these were not substantially modified by any single study. Indeed, the CMM risk estimates ranged from 1.14 (95% CI 1.07–1.24, *p* = 0.0001), omitting the study of Cervenka et al. [21], to 1.17 (95% CI 1.10–1.25, *p* < 0.0001), omitting the study of Donley et al. [24]

#### 3.3.4. Publication Bias

No significant publication bias was detected with Egger’s or Beggs method (Table 2, Figure A1).

## 4. Discussion

The incidence of cutaneous melanoma continues to increase globally [67], presenting a challenge in identifying unestablished risk factors. Melanoma is classically considered a non-hormone-related cancer; nevertheless, cutaneous melanoma has been widely investigated as a steroid hormone-sensitive cancer (particularly estrogens) [68]. Indeed, female hormones can contribute to modulate cellular proliferation and cell cycle progression through receptor-mediated transcriptional mechanisms [69]; moreover, previous studies reported the expression of progesterone and estrogen receptors in melanoma in various degrees [68,70]. Evidence suggests that estrogens may contribute to the gender differences in the immune pathways [71] and response [72,73], even though the role of sex hormones in the immunologic escape of cancer remains unclear [74,75]. Steroid hormones such as estrogen act through their cognate receptors, i.e., estrogen receptor alfa (ERα) and estrogen receptor beta (ERβ) [76]. ERs belong to the nuclear receptor superfamily, which act as transcription factors. Estrogen binding to the nuclear receptors is responsible for a nuclear translocation, with the consequent activation of genomic pathways and the transcription of multiple target genes. ERα promotes DNA transcription, while ERβ inhibits it; ERα plays a role in tumorigenesis by stimulating cell proliferation, while ERβ seems to have a significant antitumor activity [77,78]. When ERs are linked to the G protein of cellular membrane molecules, i.e., the G-protein-coupled estrogen receptor (GPER), ERs act as membrane receptors via a “non-genomic pathway”. GPER are responsible for changes in the cytosolic signaling, leading to increased activity of the RAS/BRAF/MEK axis. The GPER acts via intracellular cAMP-protein kinase (PK) and cAMP-response element-binding protein (CREB) phosphorylation. GPERs are involved in the development and progression of different cancer types. In skin, GPERs regulate melanin production and are expressed in melanoma cells. They promote melanogenesis and regulate melanocyte growth, differentiation, and function [78,79]. In conclusion, the correlation between endogenous female hormones and cutaneous melanoma has been extensively studied [49,80,81,82,83], while the potential link between exogenous female hormones, either OCs or MHT, and CMM development has only been recently investigated. This underlines the importance of investigating the influence of different types of exogenous hormones and the risk of developing CMM.

Our systematic review and meta-analysis summarized the evidence and investigated the effect of exogenous hormones on the risk of developing melanoma in women. Our analysis showed no significant association between OC use and the risk of developing CMM, and our findings agree with threeprevious meta-analyses [57,84,85].

The use of the exogenous hormone, in accordance with our meta-analysis, does not affect the risk of developing SSM or NM, even if it should be considered that this result could be influenced by a small number of studies included for SSM and NM.

Our results for OC use and the risk of developing CMM are in accordance with the recent meta-analysis of Sun et al., 2020, which is based on twenty-seven studies [85]; however, it included even letters to the editor and excluded two case–control studies [29,40] and a large cohort study [20], which was considered within our review instead.

In contrast to the previous meta-analysis by Gandini et al. [57], our meta-analysis showed a significant association between MHT and an increased risk of CMM. It is noteworthy that our meta-analysis included two multicentric studies and five cohort studies, which were excluded by Gandini et al. [57]. Our results on MHT use and the risk of developing CMM are in accordance with the two most up-to-date meta-analyses [85,86]; in fact, both suggested that the use of MHT is related with an increased risk of developing melanoma in women. In particular, our results are in accordance with Sun’s (2020) and Tang’s (2020) results in relation to hormone type (estrogen), and with Sun’s (2020) results in relation to study type (cohort). However, these two meta-analyses [85,86], as already described in two different letter to the editor [87,88], do not include three large cohort studies [19,20,48] considered within our meta-analysis. Stratifying the analysis by study design demonstrated that the increased risk of CMM in association with MHT was confirmed among prospective cohort studies, which are less prone to bias compared with retrospective studies. The type of MHT, the route of administration, and the current status of use seemed to play a role in increasing the risk of developing CMM. Our findings referring to the type of MHT suggest that exogenous estrogen presents a risk factor for CMM, while the formulations of MHT containing estrogen and progestin showed no significant association with the risk of developing CMM.

### Limitations

We are aware that our analysis has several limitations and that caution is needed in interpreting our findings. Firstly, we could not investigate the OC formulations, which differed considerably during the years of publication of the included studies. The meta- analysis on OC use and the risk of developing CMM included 12 studies published in the 1980s, 12 studies published in the 1990s, 8 studies published in the 2000s, and 6 studies published in the 2010s. Secondly, we found substantial heterogeneity among the studies, despite the availability of many relevant papers. Thirdly, the observed association between MHT and CMM risk could be partially due to unmeasured or residual confounding, although the majority of the selected studies reported risk estimates adjusted for major potential confounders (e.g., age, body mass index, smoking, pigmentary traits, and parity). Furthermore, the stratified analyses on the type of MHT, the route of administration, and the status of use were performed on a small number of risk estimates. We did not stratify the results by age, which represents a major confounding factor for the association between hormonal/reproductive factors and cancer risk. Lastly, all the included studies reported a risk estimation for CMM in Western populations. Ethnic differences are not only potentially related to pigmentary traits, but also to differences in the use of OCs or MHT, contributing to risk effects associated with CMM.

More studies are needed to further investigate the potential role of MHT or OC formulation, the duration of use, the dosage of use, the age at first and last use, as well as the cancer receptor subtype [80,86].

## 5. Conclusions

In summary, our meta-analysis showed an increased risk of CMM in women receiving MHT, while no significant association between OC use and risk of developing CMM was found. The role of exogenous hormones in CMM tumorigenesis remains controversial. Further studies are needed to investigate the potential correlations of the dosage, duration of use, and formulation of OCs and MHT with risk of CMM, and to better understand the role of potential confounders, including age at first and last use and ethnicity.

## Figures and Tables

**Figure 1 cancers-14-03192-f001:**
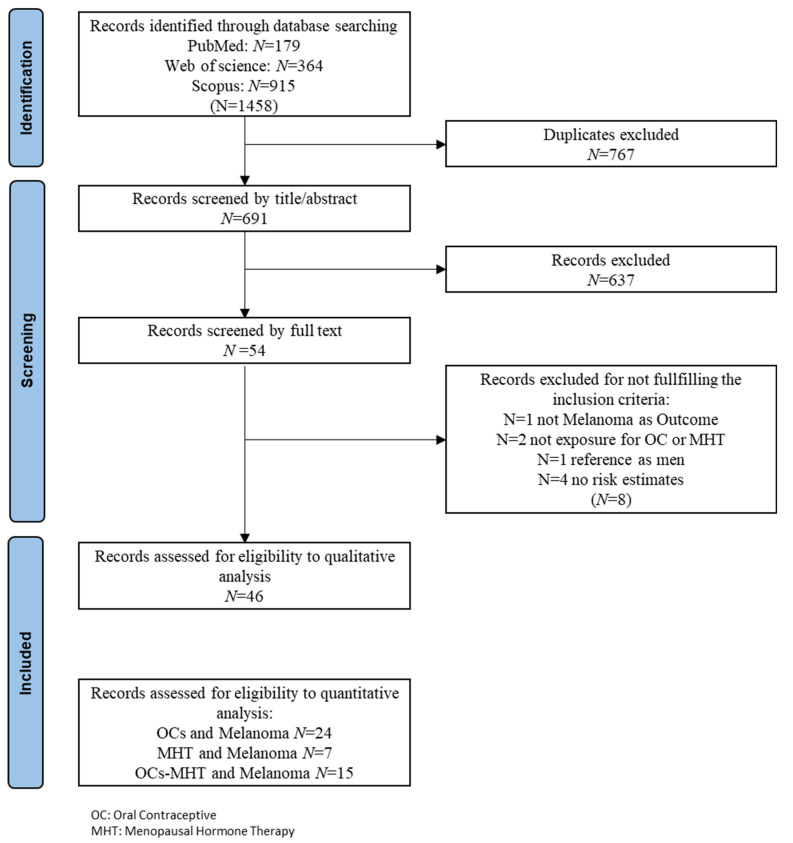
PRISMA flow chart of included studies.

**Figure 2 cancers-14-03192-f002:**
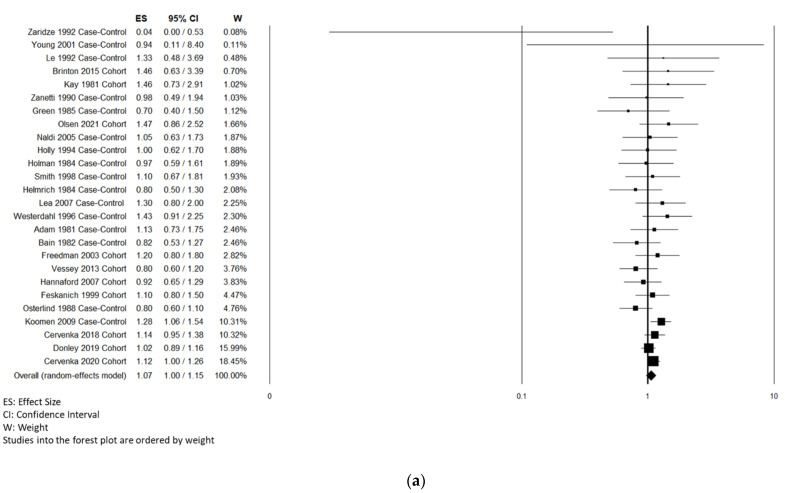
Forest plot of OC (**a**) and MHT (**b**) use and risk of CMM.

**Table 1 cancers-14-03192-t001:** Results of stratified analysis of malignant melanoma risk estimates for use of oral contraceptives (OCs). Reference category: no OC users.

	Sample Size	Combined Risk Estimate		Test of Heterogeneity			Publication Bias	
		Value (95% CI)	*p*	Q	I^2^%	*p*	P (Egger’s Test)	P (Begg’s Test)
CMM								
All (*n* = 26)	989,210	1.07 (1.00–1.15)	0.062	28.45	12.12	0.288	0.277	0.774
Study design								
Case–control (*n* = 16)	15,085	1.03 (0.89–1.19)	0.688	20.52	26.90	0.153	0.055	0.368
Cohort (*n* = 10)	974,125	1.08 (1.01–1.16)	0.032	7.90	0.00	0.544	0.572	0.421
Time since last OC use								
<4–5 years (*n* = 3)	200,907	0.91 (0.71–1.16)	0.434	2.76	0.00	0.431	0.544	1.000
5–9 years (*n* = 3)	200,907	0.90 (0.72–1.14)	0.397	0.55	0.00	0.907	0.376	0.174
>10 years (*n* = 3)	200,907	0.94 (0.58–1.51)	0.787	16.46	81.77	0.001	0.684	1.000
Status of use								
Past (*n* = 4)	334,135	1.15 (0.98–1.34)	0.086	0.13	0.00	0.988	0.768	1.000
Current (*n* = 4)	334,135	1.46 (0.95–2.25)	0.085	3.55	15.40	0.315	0.651	0.497
Age at first use								
≥20 years (*n* = 3)	389,059	1.16 (1.02–1.33)	0.028	0.41	0.00	0.817	0.652	0.602
SSM								
All (*n* = 5)	83,996	1.15 (0.83–1.59)	0.396	11.53	65.30	0.021	0.720	0.327
Study design								
Case–control (*n* = 4)	4631	1.20 (0.72–2.01)	0.489	11.31	73.48	0.010	0.823	0.497
NM								
All (*n* = 4)	83,763	0.86 (0.51–1.44)	0.569	5.16	41.89	0.160	0.975	1.000
Study design								
Case–control (*n* = 3)	4398	0.81 (0.42–1.57)	0.538	4.89	59.07	0.087	0.878	0.602
Uveal melanoma and intraocular melanoma								
All (*n* = 3)	2269	0.86 (0.64–1.15)	0.298	0.45	0.00	0.797	0.926	0.602

Abbreviations: CMM—cutaneous malignant melanoma; NM—nodular melanoma; OCs—oral contraceptives; SSM—superficial skin melanoma.

**Table 2 cancers-14-03192-t002:** Results of stratified analysis of malignant melanoma risk estimates for menopausal hormone therapy (MHT) use. Reference category: no MHT users.

	Sample Size	Combined Risk Estimate		Test of Heterogeneity			Publication Bias	
		Value (95% CI)	*p*	Q	I^2^%	*p*	*p* (Egger’s Test)	*p* (Begg’s Test)
CMM								
Ever-users (*n* = 16)	1,434,366	1.15 (1.08–1.23)	<0.001	20.09	25.32	0.169	0.972	0.719
Study design								
Case–control (*n* = 7)	182,909	1.20 (0.98–1.47)	0.077	10.44	42.52	0.107	0.969	0.881
Cohort (*n* = 9)	1,251,457	1.12 (1.04–1.19)	0.001	7.67	0.00	0.467	0.169	0.677
Duration of use								
<5 years (*n* = 7)	405,704	1.10 (0.92–1.31)	0.285	10.34	41.95	0.111	0.614	0.881
>5 years (*n* = 7)	399,472	1.06 (0.96–1.17)	0.267	0.78	0.00	0.993	0.422	0.652
Status of MHT use								
Current users (*n* = 4)	975,710	1.19 (1.09–1.30)	0.001	0.25	0.00	0.970	0.327	0.497
Past users (*n* = 4)	975,710	1.09 (0.89–1.33)	0.418	0.36	67.95	0.025	0.134	0.174
Route of administration								
Oral (*n* = 3)	384,140	1.19 (1.11–1.27)	<0.001	1.34	0.00	0.511	0.224	0.602
Transdermal–cutaneous (*n* = 3)	384,140	1.36 (1.19–1.54)	<0.001	0.12	0.00	0.941	0.112	0.117
Type of MHT								
ET (*n* = 6)	999,574	1.34 (1.18–1.52)	<0.001	3.45	0.00	0.632	0.280	0.851
EPT (*n* = 5)	976,330	1.12 (0.97–1.30)	0.119	6.16	35.11	0.187	0.457	0.624
Uveal melanoma and intraocular melanoma								
All (*n* = 3)	2269	1.32 (0.75–2.33)	0.328	7.57	73.59	0.023	0.654	0.602

Abbreviations: CMM—cutaneous malignant melanoma; ET—estrogen therapy; EPT—estrogen–progestin therapy; MHT—menopausal hormone therapy.

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
