# Peer review of "Exogenous Hormone Factors in Relation to the Risk of Malignant Melanoma in Women: A Systematic Review and Meta-Analysis"

_cancers, 2022, doi:10.3390/cancers14133192_

Round 1
Reviewer 1 Report
The manuscript discusses the impact of exogenous use of hormones on risk of cutaneous and uveal melanoma. The meta-analysis was addressed properly.
Meta-regression is suggested.
Tables 1 and 2. please add a column for the sample size used for each pooled results.
Appendix A: outlay of the table which is almost 90 pages is confusing and hard to follow the content or realize comparisons between studies. It could be split into multiple tables with consistency within the cells.
Appendix C1: x axis need to be plotted
Author Response
Thank you very much for all your suggestion, we have attach a file with a point-by-point response to the reviewer.

Reviewer 2 Report
Thank you for this detailed systemic review and meta-analysis on the effects of exogenous hormone use on melanoma development. The manuscript represents a significant amount of work or a controversial topic with conflicting evidence in the literature. Several items need to be addressed prior to publication of the study.
1. The authors use a confusing categorization of melanoma (CMM, superficial spreading, nodular, and uveal). It is not entirely clear what CMM refers to in this context, as it typically includes superficial spreading, nodular, and melanoma NOS. In fact, melanoma NOS typically represents the majority of cutaneous melanomas and if substratification is to be utilized in this study, its absence as a category is problematic. Additionally, uveal melanomas do not typically share the same risk factors as cutaneous melanoma and its inclusion in this study distracts from the focus of the manuscript. If uveal melanoma is to be included, then efforts should also be made to include other rare variants, such as mucosal and aural lentiginous melanomas. Otherwise, the manuscript seems to be picking a choosing what it reports without appropriate justification.
2. It is not entirely clear from the methods section as to what variables were included in the random effects models used in the meta-analyses. Were these adjusted for age, race/ethnicity, medical comorbidities/immunosuppression, history of UV exposure. The latter is particularly important as tanning bed use is likely to be a significant factor contributing to melanomas among younger women and may be largely responsible for the significant associations observed in this cohort.
3. On page 5, lines 182-183, the authors make reference to a study by Hannaford, which collected outcome information from GP records in one cohort and from hospital discharge records in the other one. This seems to be a significant source of bias and this study should either be excluded from the manuscript or explicitly removed in a sensitivity analysis.
4. On page 6, the authors report that exogenous hormone use did not affect risk of SSM or nodular melanoma. As a result, it seems that only risk of CMM (which is vaguely defined by the authors) is affected. No attempt is made to explain these observations, if they are true and not entirely caused by the small number of included studies for SSM and NM.
5. It is important to note that most of the ORs reported in this manuscript are not in the clinically actionable window (typically in the 1.5-2.0 fold increased risk range), even if they are statistically significant.
6. On page 12, lines 245-246, the authors remove one study at a time, but note that removal of 2 specific studies heavily affects the meta-analysis results. Since these are both outliers, efforts should be made to remove both outliers at the same time and report on the results as a sensitivity analysis.
Author Response

(The authors gave the same response as above.)

Reviewer 3 Report
The authors described "Exogenous Hormone Factors in Relation to the Risk of Malignant Melanoma in Women: A Systematic Review and Meta Analysis. As they pointed out, cutaneous melanoma is classically considered a non-hormone-related cancer, nevertheless cutaneous melanoma has widely investigated as a steroid hormone-sensitive cancer. Their systematic review and meta-analysis summarized the evidence and investigated the effect of exogenous hormone on the risk of CMM risk in women. Therefore, this study should be attractive for potential readers, especially in oncologists. I have no questions and suggestions for this amazing study. Thank you for this opportunity.
Author Response
Thank you very much for your evaluation.
Round 2
Reviewer 1 Report
Sample size in the table is fracture, can you explain please?
Author Response
Dear Reviewer,
I do not want to hide that we have struggled to understand what your question meant. I am telling you this to anticipate that maybe what we did was not what you request. Probably this difficulty is due to differente meaning of the use of "fracture" in ours languages. We have attached here a revised versione of the manuscript, where you can find that commas have replaced dots in the sample sizes. This should make simple sizes more readable.
We remain available. Thanks again for your suggestion.
